# Multi-Trajectories of Macronutrient Intake and Their Associations with Obesity among Chinese Adults from 1991 to 2018: A Prospective Study

**DOI:** 10.3390/nu14010013

**Published:** 2021-12-21

**Authors:** Xiaofan Zhang, Jiguo Zhang, Wenwen Du, Chang Su, Yifei Ouyang, Feifei Huang, Xiaofang Jia, Li Li, Jing Bai, Bing Zhang, Zhihong Wang, Shufa Du, Huijun Wang

**Affiliations:** 1Chinese Center for Disease Control and Prevention, National Institute for Nutrition and Health, Beijing 100050, China; zhangxf@ninh.chinacdc.cn (X.Z.); zhangjg@ninh.chinacdc.cn (J.Z.); duww@ninh.chinacdc.cn (W.D.); suchang@ninh.chinacdc.cn (C.S.); ouyyf@ninh.chinacdc.cn (Y.O.); huangff@ninh.chinacdc.cn (F.H.); jiaxf@ninh.chinacdc.cn (X.J.); lili@ninh.chinacdc.cn (L.L.); baijing@ninh.chinacdc.cn (J.B.); zhangbing@chinacdc.cn (B.Z.); wangzh@ninh.chinacdc.cn (Z.W.); 2Department of Nutrition and Carolina Population Center, University of North Carolina at Chapel Hill, Chapel Hill, NC 27599, USA; dushufa@unc.edu

**Keywords:** macronutrient, multi-trajectories, obesity, prospective study, China

## Abstract

Studies on macronutrient intake and obesity have been inconclusive. This study examined the associations between multi-trajectories of macronutrients and the risk of obesity in China. We used data from 7914 adults who participated in the China Health and Nutrition Survey at least three times from 1991 to 2018. We collected detailed dietary data by conducting three 24 h dietary recalls and weighing foods and condiments in household inventories. We identified multi-trajectories using group-based multi-trajectory models and examined their associations with the risk of obesity with multiple Cox regression models. We found four multi-trajectories in rural areas: balanced macronutrient intake (BM), moderate protein, increasing low fat, and decreasing high carbohydrate (MP&ILF&DHC); decreasing moderate protein, decreasing high fat, and increasing moderate carbohydrate (DMP&DHF&IMC); increasing moderate protein, increasing high fat, and decreasing low carbohydrate (IMP&IHF&DLC)—35.1%, 21.3%, 20.1%, and 23.5% of our rural participant population, respectively. Compared with the BM trajectory, the hazard ratios of obesity in the DMP&DHF&IMC and the IMP&IHF&DLC groups were 0.50 (95% confidence interval (CI): 0.27–0.95) and 0.48 (95% CI: 0.28–0.83), respectively, in rural participants. Relatively low carbohydrate and high fat intakes with complementary dynamic trends are associated with a lower risk of obesity in rural Chinese adults.

## 1. Introduction

The World Health Organization (WHO) recently estimated that obesity has nearly tripled worldwide since 1975 and more than 650 million adults were obese in 2016 [1]. This issue has grown to epidemic proportions. According to the Global Burden of Disease 2015 Obesity Collaborators, over 4 million people died as a consequence of being overweight or obese [2]. Once considered problems of high-income countries, overweight and obesity are now on the rise in low- and middle-income countries. In China, obesity in adults has increased steadily since the early 1980s. An estimated 85 million adults aged 18–69 years in China were obese in 2018, three times the number in 2004 [3]. 

Nutrition is the primary modifiable factor that can counter the rising prevalence of obesity. The overall macronutrient composition of the diet, rather than merely calorie intake, may influence body weight. However, the association between macronutrient intake (defined as the proportions of energy derived from carbohydrates, proteins, and fats) and obesity is a matter of active debate. Some authors point out that the low-fat (LF), high-carbohydrate (HC) diets that the US government recommended in the 1970s led to the rise in obesity throughout the 1980s and 1990s [4]. A meta-analysis of randomized trials among overweight/obese adults showed that low-carbohydrate (LC) (≤40%) and LF (≤20%) diets had similar effects on weight loss [5]. Other studies reported similar results [6,7]. However, some studies have shown that an LC diet is more effective in weight loss than an LF diet [8,9]. Most studies of healthy people indicate that high protein (HP) intake is associated with an increase in weight [10,11,12]. By contrast, intervention studies in obese/overweight persons usually find that an HP diet is effective in achieving and, to some extent, maintaining weight loss [13]. A two-year intervention trial found that reduced-calorie diets result in clinically meaningful weight loss regardless of which macronutrients they emphasize [14]. 

Most studies of the relationship between macronutrient intake and obesity have been short-term trials in a small sample of obese European populations, and the results have been inconsistent [4,5,6,7,8,9,10,11,12,13,14]. The Chinese dietary pattern is very different from that of Europeans. Chinese dietary macronutrient intakes are not static but show different trends. For instance, the trend in the estimated percentage of energy intake from fat in China spiked from 1982 to 2012 [15]. Researchers have not examined the effects of this dynamic dietary macronutrient trend on obesity in non-overweight and non-obese Chinese adults. To fill this gap in the literature, we used data from 1991 to 2018 in the China Health and Nutrition Survey (CHNS) to describe different multi-trajectories of macronutrient intakes and examined the associations between these trajectories and the risks of obesity in China.

## 2. Materials and Methods

### 2.1. Study Participants

The CHNS is an ongoing longitudinal survey designed to investigate health and nutritional status in Chinese populations. Its study design and procedures have been described previously [16]. The CHNS completed rounds in 1989, 1991, 1993, 1997, 2000, 2004, 2006, 2009, 2011, 2015, and 2018. In the current analyses, we used all rounds of CHNS survey data except 1989. Our sample included adults over 18 years old in 10 surveys between 1991 and 2018. We further analyzed the relationships between the multi-trajectories of macronutrient energy supply ratios and new-onset obesity. By the end of 2018, the CHNS had 42,736 participants. We excluded 9367 participants who were younger than 18 years; 360 pregnant or nursing mothers; 1523 participants without dietary surveys or physical measurements; 226 participants with abnormal data on energy intakes (male >6000 kilocalories (kcal) or <800 kcal, female >4000 kcal or <600 kcal) or physical measurements (height <120 cm (cm) or body mass index (BMI) < 15 kg per meter squared (kg/m^2^) or >40 kg/m^2^); 8576 participants with overweight or obesity (BMI ≥ 24 kg/m^2^); and 14,770 people who participated in only one or two surveys (Figure 1). The number of visits ranged from 3.0 to 10.0 per participant: 3.0 visits, number of participants (*n*) = 2633; 4.0 visits, *n* = 1533; 5.0 visits, *n* = 1049; 6.0 visits, *n* = 908; 7.0 visits, *n* = 707; 8.0 visits, *n* = 536; 9.0 visits, *n* = 338; 10.0 visits, *n* = 210 (median 5.7 visits). Our study included 7914 participants and 39,103 observations. 

The Institutional Review Board of the University of North Carolina at Chapel Hill, Chapel Hill, North Carolina, United States (No. 07-1963), and the Institutional Review Committee of the National Institute for Nutrition and Health, Chinese Center for Disease Control and Prevention, Beijing, China, approved the survey protocols and instruments and the process for obtaining informed consent (No. 201524). All participants provided written informed consent prior to the surveys.

### 2.2. Outcome Variables

Obesity was our major outcome variable. Based on standard protocols recommended by the WHO [17], in each survey, our trained physicians and nurses measured height without shoes to the nearest 0.1 cm using a portable SECA stadiometer (SECA, Hamburg, Germany), and they measured body weight without shoes and with light clothing to the nearest 0.1 kg (kg) using a calibrated beam scale. We calculated BMI as weight in kg divided by height in meters (m) squared (kg/m^2^) and defined obesity as a BMI ≥ 28.0 kg/m^2^ according to the Chinese adult weight criteria (WS/T 428-2013).

### 2.3. Exposure Variables

The field work of each round was launched in autumn, considering the comparability between survey rounds. Percentages of energy intake from proteins, fats, and carbohydrates were our primary exposure variables, which we analyzed as a whole. To collect detailed dietary data, we (1) conducted three consecutive 24 h recalls (one weekend day and two weekdays) for all foods, snacks, and beverages consumed at home or away from home at the individual level over the previous 24 h for each of the three days; (2) weighed food items and condiments (e.g., edible oils and salt) added during food preparation and cooking at the household level over the same three-day period; and (3) used food pictures, food samples, and a food diary to help complete 24 h recalls. Interviewers weighed and recorded all foods remaining after the last meal of the survey. When food was discarded and weighing was not possible, we used food pictures to help estimate the amount of food discarded. We calculated total energy intake per day in kilocalories (kcal/d) and our primary exposure variables based on the diet data that we collected and the Chinese Food Composition Table. Calculations of energy derived from carbohydrate, protein, and fat employed the standard conversion factors for grams to kilocalories (4 kcal/g for carbohydrate and protein, 9 kcal/g for fat) and the proportions of energy derived from macronutrients were then calculated as energies imparted by a particular macronutrient/total energy intake (%).

This combination of three-day 24 h recalls, household weighing, and food pictures and food diaries can improve the accuracy of recalls [18]. All interviewers participated in at least one seven-day training session and passed a comprehensive test before collecting any data. For each dietary recall day, investigators went to participants’ homes and helped to record food intake during the past 24 h. Investigators also weighed the household cooking oil and condiments at the beginning and end of each 24 h dietary survey. Detailed dietary data collection and allocation have been described elsewhere [19,20].

### 2.4. Covariates

In our analysis, we included a number of confounders related to diet and obesity, including age, residing in the north, education level, marital status, family income, smoking status, alcohol drinking status, fruit intake, vegetable intake, physical activity (PA), sedentary time (ST), sleep time, baseline BMI, waist circumference (WC), total energy intake, and disease history. Due to the huge disparity in development between urban and rural areas in China, the eating behaviors and obesity rates of the residents are also quite different. We separated residency into two groups: urban areas, including communities in large cities and county capital cities; and rural areas, including communities in highly rural suburban areas and rural villages. The line between the Qinling Mountains and the Huai River divides China into the north and the south. In this study, we consider Beijing, Hebei, Liaoning, Heilongjiang, Shandong, Henan, and Shaanxi provinces as the north and the remaining provinces as the south. We collected detailed data on family incomes from all sources, including wages, farming, gardening, household businesses, and other income sources, and inflated the incomes to the value of the 2018 Chinese renminbi (RMB). We classified smokers as participants who had smoked at least one cigarette per day currently or formerly and defined alcohol drinkers as participants who drank beer or other types of alcohol last year. Previous studies showed that sleep duration was associated with risk of obesity [21,22], so we also asked participants how many hours of sleep they obtained each day. A history of illness was based on whether a doctor had made a diagnosis or not. We included four PA domains—occupational, household, leisure time, and transportation—and four ST domains—leisure and television time, computer time, reading time, and game time. Participants reported their average hours per week (hours/week) of all types of PA and ST during the past year. We calculated the total metabolic equivalents of tasks (METs) of PA as MET hours per week (METs/week). We measured WC in cm midway between the lowest rib margin and the top of the iliac crest. 

### 2.5. Statistical Analysis

We used group-based multi-trajectory modeling to determine the multi-trajectories of macronutrient energy supply ratios (hereinafter referred to as “macronutrient trajectories”) divided into urban and rural areas [23]. We constructed the model as a function of age to describe the probability of adhering to a percentage of energy intake from proteins (Protein%), a percentage of energy intake from fats (Fat%), and a percentage of energy intake from carbohydrates (Carbohydrate%) concurrently over time. We performed multi-trajectory modeling with a STATA plug-in using continuous norming (cNORM) distribution for continuous data [24]. We used age as a timescale for the trajectories. We considered linear, quadratic, and cubic terms of age and evaluated them based on their significance levels. We tested participants in two, three, four, five, and six trajectory groups for linear, quadratic, and cubic specifications for trajectory shape until we established the best-fitting model. We used statistically rigorous criteria to determine the best fit. (1) With the lowest Bayesian information criterion (BIC), we used the magnitude of difference in BIC (percentage change) to choose between more complex (with one additional specified trajectory group) and simpler models; (2) we included at least 2% of the sample population in each trajectory class; and (3) we ascertained within each group values of average posterior probability of membership in which values greater than 0.7 indicate adequate internal reliability [25]. Once we determined the macronutrient trajectories, we created a nominal categorical variable to describe the trajectory class of each participant and then used this in Cox multivariate regression models.

We used Cox multivariate regression models with age as the timescale to estimate associations between macronutrient trajectories and risk of obesity. We calculated hazard ratios (HRs) and 95% confidence intervals (CIs). Time at entry was the participant’s age at baseline, and exit time was the age at which the participant was diagnosed with obesity, was lost to follow-up, or was censored at the end of the follow-up period, whichever came first. We adjusted models for age, gender, residence in north or south region, education, family income, alcohol consumption, cigarette smoking, fruit intake, vegetable intake, PA, ST, sleep time, baseline BMI, WC, energy intake, and disease history. We performed statistical analyses using SAS 9.4 (SAS Institute Inc., Cary, NC, USA). We considered a two-sided *p* value < 0.05 statistically significant.

## 3. Results

### 3.1. Sample Characteristics 

Following the initial screening process, our study included 7914 participants (BMIs < 24 kg/m^2^), all of whom were visited three or more times from 1991 to 2018 (Figure 1). The mean age of the participants ranged from 42.1 in 1991 to 58.8 in 2018, with a basic gender balance; more than two thirds were rural residents, and more than 60.0% were southerners. Education level and annual household income increased with the survey year, while the proportion of smokers and drinkers showed a decreasing trend during the survey years. The amount of PA decreased significantly, while ST increased slowly. In terms of the percentages of energy intake from macronutrients, the percentage of protein intake did not change much, the percentage of fat intake increased from 24.1% to 34.7%, and the percentage of carbohydrate intake decreased from 62.7% to 51.5%. The total energy intake decreased, while BMI and WC increased (Table 1).

### 3.2. Multi-Trajectories of Macronutrient Energy Supply Ratios

In the total sample of 10 waves of CHNS data, we identified four distinct multi-trajectories of percentages of energy intakes from proteins, fats, and carbohydrates in both urban and rural areas (Figure 2). We named the trajectories based on the appropriate ranges of protein intake, 10–15%, fat intake, 20–30%, and carbohydrate intake, 50–65%, from the Chinese Dietary Reference Intakes (2013) [26]. Our sample included 2586 residents in urban areas and 5328 in rural areas.

Among urban residents the first group, which we identified as increasing moderate protein (IMP), high fat (HF), and low carbohydrate (LC) (IMP&HF&LC), included 38.7% of the urban participant population. These persons’ percentage of energy intake from proteins was over 14.0% and slowly increased over time, that from fats was around 35.0%, and that from carbohydrates was less than 50.0%. The second group, with almost balanced macronutrient intake (ABM), amounted to 33.1% of the urban participants and included persons who maintained protein intakes at approximately 12.0%, whose fat intakes slightly exceeded 30.0% and increased over time, and whose moderate carbohydrate intakes were between 55.0% and 60.0% and decreased over time. The third group, with moderate protein (MP), very high fat (VHF), and very low carbohydrate (VLC) (MP&VHF&VLC), amounting to 19.7% of the urban participants, included persons with protein intakes at approximately 13.0%, fat intakes at more than 40.0%, and carbohydrate intakes as low as 40%. The fourth group, with moderate protein (MP), increasing low fat (ILF), and decreasing high carbohydrate (DHC), (MP&ILF&DHC), amounted to 8.5% of the urban participants and included persons whose protein intakes were moderate, between 11.0% and 12.0%, whose fat intakes gradually approached 20.0%, and whose carbohydrate intakes slowly decreased from more than 70.0% to more than 65.0%.

Among rural residents, the first group, who had balanced macronutrient intake (BM), included 35.1% of the rural participant population. This group’s percentage of energy intake from proteins was between 11.0% and 12.0%, that from fats gradually increased from 20.0% to 30.0%, and that from carbohydrates slowly decreased from 65.0% to 55.0%. All macronutrient energy intake percentages in this group were within the appropriate ranges for Chinese adults [21]. The second group, with moderate protein (MP), increasing low fat (ILF), and decreasing high carbohydrate (DHC) (MP&ILF&DHC), amounted to 21.3% of the rural participants and was similar to the fourth urban group (8.5% of the urban participants). The third group, with decreasing moderate protein (DMP), decreasing high fat (DHF), and increasing moderate carbohydrate (IMC) (DMP&DHF&IMC), amounted to 20.1% of the rural participants and included persons with protein intakes that ranged from 12.0% to 13.0% and decreased slowly over time, fat intakes that gradually decreased from more than 30.0% to less than 30.0%, and carbohydrate intakes that gradually increased from 50.0% to more than 60.0%. The fourth group, with increasing moderate protein (IMP), increasing high fat (IHF), and decreasing low carbohydrate (DLC) (IMP&IHF&DLC), included 23.5% of the rural participants and included persons with protein intakes that slowly increased from 12.0% to 13.0%, fat intakes that gradually increased from 30.0% to 40.0%, and carbohydrate intakes that decreased from 50.0% to 40.0%.

### 3.3. Baseline Characteristics by Multi-Trajectories

Table 2 details the characteristics of the persons included in each trajectory. The urban IMP&HF&LC group had the largest population, 38.7%, while the majority of rural residents were in the BM group (35.1%). 

Compared with other urban trajectory groups, the MP&VHF&VLC group had the highest fat intake level (42.2% on average), the lowest carbohydrate intake level (43.9% on average), and the highest total energy intake (2408.8 kcal on average). In terms of population characteristics, this group tended to be young people in southern areas with higher education levels and family incomes. 

Compared with other rural trajectory groups, the DMP&DHF&IMC and IMP&IHF&DLC groups had similar characteristics, with moderate protein intakes, higher fat intakes, and lower carbohydrate intakes, but the trajectories changed in opposite directions. These groups tended to be young people in southern areas with higher education levels and family incomes and with lower rates of smoking and alcohol consumption and total energy intakes.

### 3.4. Associations between Multi-Trajectories and Obesity

Table 3 presents the findings of the Cox multivariate regression analyses exploring associations between multi-trajectories and risk of obesity. Compared with the ABM trajectory, the three other urban groups were not associated with obesity risk. Compared with the BM trajectory, the DMP&DHF&IMC (HR: 0.50, 95% CI: 0.27–0.95) and IMP&IHF&DLC (HR: 0.48, 95% CI: 0.28–0.83) rural trajectories were significantly associated with decreased risks of obesity when adjusted for all covariates (age, gender, residence in north or south region, education, family income, alcohol consumption, cigarette smoking, fruit intake, vegetable intake, PA, ST, sleep time, baseline BMI, WC, total energy intake, and disease history).

## 4. Discussion

Using 10 waves of data from the CHNS, a nationwide cohort study, from 1991 to 2018, this study examined the trends in macronutrient intake (proteins, fats, and carbohydrates) as a whole and their multi-trajectory associations with obesity among Chinese adults. Similar to the national general trend [15], we found that the total energy intake of our sample population with no overweight or obesity decreased from 2423.9 kcal in 1991 to 1971.1 kcal in 2018. The percentage of energy intake from fats increased from 24.1% to 34.7%, that from carbohydrates decreased from 62.7% to 51.5%, and that from proteins stabilized within a small range. This differs from other countries. A cross-sectional analysis of the US using nationally representative data in 1999–2016 showed a decrease from 52.5% to 50.5% in energy intake from carbohydrates, whereas energy intakes from proteins and fats increased from 15.5% to 16.4% and from 32.0% to 33.2%, respectively [27]. Macronutrient trends from national nutrition surveys in Australia illustrated another result [28]. Given that the trends of macronutrient intakes in China are different from those in other countries, it is necessary to explore the relationship between the trajectories of macronutrient changes and the risk of obesity in the Chinese population.

To date, we have not found any longitudinal study that determines multi-trajectories of individual macronutrient intakes. The trajectories that we identified in this study provide new insights for the common progression of macronutrient intakes that could be expected to be observed in non-overweight and non-obese adults. When grouping the long-term trend of macronutrient intake, we found that the multi-trajectories of urban and rural macronutrient intakes were very different. On the whole, the macronutrient intakes in rural areas were closer to the recommended amounts, and the trends of multi-trajectory changes with time were more obvious, with more adults in the LF and HC groups. City dwellers, on the contrary, consumed higher percentages of fat. First, the multi-trajectories of urban participants were relatively stable on the whole, while rural participants had more obvious trends of rising or falling over time. Second, we found no BM group in urban areas. The composition of the three macronutrients in the urban ABM group was closest to the recommended range, but the fat intake in the ABM group increased slowly over time and exceeded the upper recommended limit of 30.0%. In rural areas, the number of people in the BM group accounted for the largest proportion, 35.1%. Third, the level of fat intake among urban residents was higher than that among rural residents. In urban areas, 19.7% of the residents acquired more than 40.0% of their energy from fats, while, among all rural residents, the percentage from fats was lower than 40.0%. Moreover, the proportion of people in urban areas with a high fat intake (>30.0%) was 58.4%, compared to 43.6% in rural areas. Fourth, the proportion of people in the LF and HC groups in rural areas was 21.3%, compared with 8.55% in urban areas.

Our study found that the multi-trajectories of macronutrient intakes in urban and rural areas were very different. A relevant longitudinal study in China also found persistent urban–rural disparities in macronutrient consumption from 1991 to 2015 [29]. The difference may be related to income inequality. Another study showed that high proportions of energy intakes from fats (>30%) were more concentrated among the rich, whereas carbohydrate intakes were larger among the poor [30]. This is similar to the results of our study, whereby urban residents had higher fat intakes and rural residents had higher carbohydrate intakes. In recent decades, China has experienced tremendous economic growth and also growing socioeconomic-related health inequalities between urban and rural areas [31]. Due to the large differences in these trajectories, we explored the associations between multi-trajectories of macronutrient intake and obesity risk in urban and rural adults.

Our analysis of CHNS data found that both the DMP&DHF&IMC and the IMP&IHF&DLC rural groups were associated with a lower risk of obesity. In the DMP&DHF&IMC group, energy intakes from fats decreased from 35% to 25%, and energy intakes from carbohydrates increased from 50% to 60%. In the IMP&IHF&DLC group, energy intakes from fats increased from 30% to 40%, and energy intakes from carbohydrates decreased from 50% to 40%. Overall, carbohydrate intakes in both groups were near or below the lower limit of appropriate intake (50%), while fat intake was near or above the upper limit (30%). This means that, for healthy adults, relatively low carbohydrate and high fat intakes may be beneficial for preventing obesity. However, it is not simply that low carbohydrate and high fat intakes are effective. A complementary dynamic is also important. When carbohydrate intake gradually increases from the lower limit, fat intake needs to gradually approach the lower limit, and when fat intake exceeds the upper limit, carbohydrate intake needs to fall below the lower limit to be effective in protecting from obesity. 

The urban IMP&HF&LC group had no protection against obesity. This may be related to the fact that the fats that urban residents consume tend to be mainly saturated fats, whereas the fats that rural residents consume tend to be plant-based cooking oils, i.e., unsaturated fatty acids. In addition, the amount of fat exceeding the upper limit should not be large. For example, the urban MP&VHF&VLC group had no protection against obesity, because the fat levels consistently exceeded 40% even though carbohydrate levels were consistently low. This lack of protection may be due to the obesity-promoting effect of very high fat levels.

Other studies have found that low carbohydrate intakes have a protective effect against obesity, but most of them examined obese people who restricted their carbohydrate intakes [32,33,34,35]. Proponents of an LC diet argue that it can increase energy expenditure by 400 to 600 kcal/d, equivalent to an hour of moderate PA. Moreover, compared with an isocaloric, higher-carbohydrate diet, the LC diet has been purported to result in preferential body fat loss due to reduced insulin secretion, promoted fat decomposition and oxidation, and reduced fat synthesis [36,37]. Another mechanism of an LC diet is its ability to spontaneously reduce calorie intake. LC diets tend to lead to increased protein intake, which may increase satiety and reduce total energy intake, thus reducing body fat [38]. Some researchers do not support these claims. In their studies, the extra energy expenditure associated with an LC diet was much lower than the diet’s proponents claimed, and the effect waned over time [39,40]. In another study, LC diets did not consistently suppress appetite more than other diets, and it may be that other factors that influence food intake (for example, habitual eating patterns and social pressures) dominate over the long term [41]. Additionally, the definition of LC intake varies in different studies, a critically important limitation. More research is required to better understand the long-term consequences of LC diets.

Studies of the relationship between fat intake and obesity are still contradictory. The WHO and the 2016 Dietary Guidelines for Chinese Residents recommend no more than 30% of total energy intake from fat [26,42]. Some studies have shown that a high-fat (HF) diet increases the risk of obesity and that LF diets are beneficial for weight loss [6,9,43,44]. However, the 2015–2020 Dietary Guidelines for Americans removed the limitations on dietary fat intake [45], because other studies found weight loss in people on both HF and LF diets [46,47,48]. Our research also found that an HF diet (30–40%) among adults of a healthy weight may help to reduce the risk of obesity. The inconsistency could have a number of explanations, one of which may be biological differences in the study population, some of whom are obese, insulin-resistant, or healthy. Additionally, the food source of the carbohydrate or fat also affects the ratios of saturated fats to unsaturated fats and highly processed, refined carbohydrates to natural carbohydrates, all of which have different effects on obesity [49]. 

The present study breaks new ground by exploring multi-trajectories of macronutrient intake and their associations with obesity among Chinese adults from 1991 to 2018. This is the first study of the longitudinal changes in macronutrient intakes in Chinese adults over 27 consecutive years, and we identified different trajectories of macronutrient intakes in urban and rural areas. In addition, this is the only study to date to examine longitudinal changes in macronutrient intake and obesity risk in a Chinese population. We used an innovative multi-trajectory modeling technique to incorporate the intercorrelations among the multiple macronutrients to improve the accuracy of individually specific probabilities of group membership, whereas the conventional group-based trajectory analysis clusters longitudinal trajectories based on one outcome. The findings could provide a scientific basis for government policymakers to design programs to prevent obesity among Chinese adults. Nonetheless, this study also has some limitations. First, self-reported dietary information may be subject to memory and recall issues. Second, loss to follow-up in longitudinal studies may result in bias. Third, with the rising popularity of take-out and dining out in recent years, our data may have underestimated cooking oils as a source of dietary fat.

## 5. Conclusions

In conclusion, our multi-trajectories of urban and rural macronutrient intakes are very different. In particular, almost 20% of urban adults consistently derived more than 40% of their energy from fats and around 40% from carbohydrates, which we did not see in the rural population. This suggests that the obvious differences between urban and rural areas should be taken into account when planning dietary interventions for Chinese adults. Relatively LC and HF intakes with complementary dynamic trends are associated with a lower risk of obesity in Chinese rural adults. This study’s design is different from previous short, small, weight-loss intervention studies for obese patients and also has a higher level of evidence than observational studies. We describe the optimal macronutrient ratios for healthy people of normal weight to avoid obesity, which has valuable significance for obesity prevention among the general population.

## Figures and Tables

**Figure 1 nutrients-14-00013-f001:**
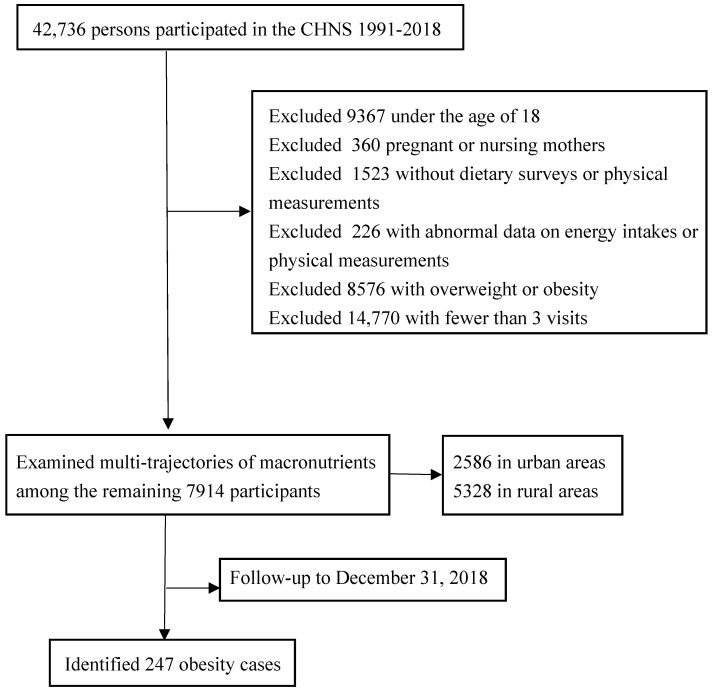
Flowchart of the participants included in the current analysis.

**Figure 2 nutrients-14-00013-f002:**
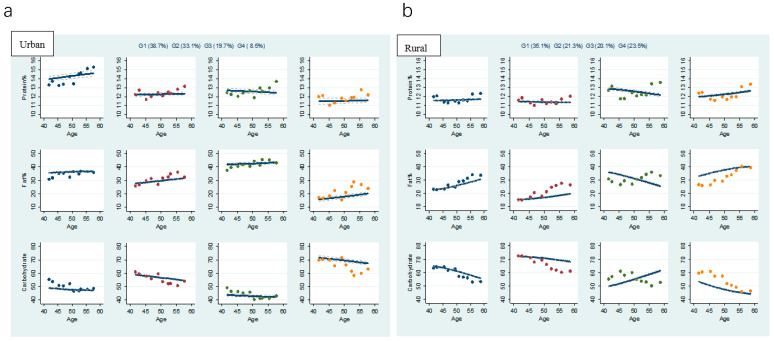
Multi-trajectories of Protein%, Fat%, and Carbohydrate% among Chinese adults by urban (**a**) or rural (**b**) residence. Source: CHNS 1991–2018. Notes: Solid lines represent the average estimated Protein%, Fat%, and Carbohydrate% (the percentage of energy provided by dietary proteins, fats, and carbohydrates) over time. Dashed lines represent the 95% CI. The dots represent the actual data, where we weighted each individual’s responses based on posterior probabilities of group membership.

**Table 1 nutrients-14-00013-t001:** Characteristics of the study population.

	Survey Year									
	1991	1993	1997	2000	2004	2006	2009	2011	2015	2018
	(*n* = 3541)	(*n* = 3732)	(*n* = 4062)	(*n* = 4414)	(*n* = 4429)	(*n* = 4314)	(*n* = 4216)	(*n* = 4183)	(*n* = 3501)	(*n* = 2711)
Age ^1^ (years)	42.1 ± 13.6	43.3 ± 14.0	45.9 ± 14.5	47.3 ± 14.5	49.7 ± 14.6	51.3 ± 14.5	52.9 ± 14.6	54.2 ± 14.6	55.3 ± 14.2	58.8 ± 13.8
Men (%)	51.2	50.6	53.0	50.7	49.2	48.6	48.3	47.1	41.9	42.8
Urban (%)	26.3	26.2	29.8	30.2	30.3	30.6	30.1	32.6	32.3	32.5
North (%)	26.7	26.9	27.3	32.9	35.2	35.3	36.5	34.2	31.3	30.1
Education (%)										
Primary and below	61.6	59.7	58.6	52.7	48.8	47.4	47.8	45.2	39.3	37.5
Junior high	25.2	26.0	26.5	28.6	30.0	29.2	31.0	29.8	33.2	33.1
Senior high and above	13.2	14.3	14.9	18.8	21.2	23.4	21.2	25.0	27.5	29.4
Annual per capita family income (%)										
Low (<10,000 RMB)	88.1	60.1	62.3	55.9	51.0	46.4	43.4	39.2	36.1	34.5
Middle (10,000–20,000 RMB)	7.5	21.7	19.4	23.0	23.6	24.7	25.5	24.6	23.9	23.2
High (>20,000 RMB)	4.3	18.2	18.3	21.1	25.4	28.9	31.0	36.2	40.0	42.3
Smoker (%)	39.2	37.7	37.4	35.5	36.1	34.7	34.3	33.4	25.7	23.1
Alcohol drinker (%)	40.1	38.0	39.0	36.3	33.5	32.9	32.5	31.9	25.7	22.6
Protein ^1^ (%)	12.2 ± 2.5	12.4 ± 2.7	11.6 ± 2.4	11.7 ± 2.5	12.2 ± 2.8	11.8 ± 2.7	12.2 ± 2.9	12.2 ± 3.1	13.0 ± 3.3	13.2 ± 3.3
Fat^1^ (%)	24.1± 11.9	23.9 ± 12.4	25.4 ± 11.9	28.3 ± 11.1	26.3 ± 12.2	30.4 ± 11.8	31.4 ± 10.6	34.2 ± 11.9	35.9 ± 12.3	34.7 ± 12.0
Carbohydrate ^1^ (%)	62.7 ± 13.0	62.7 ± 13.4	62.2 ± 12.4	59.2 ± 11.8	60.6 ± 12.7	55.1 ± 13.4	53.7 ± 12.0	52.7 ± 12.1	50.4 ± 12.5	51.5 ± 12.3
Energy intake ^1^ (kcal/d)	2423.9 ± 700.5	2380.3 ± 673.0	2486.1 ± 737.5	2371.6 ± 690.0	2303.1 ± 720.9	2315.5 ± 737.1	2194.9 ± 678.6	2060.5 ± 706.4	1990.6 ± 690.7	1971.1 ± 656.9
PA ^2^ (METs/week)	488.3 (257.5, 660.4)	373.1 (217.8, 550.0)	368.0 (164.6, 534.2)	273.8 (122.7, 435.6)	164.5 (62.1, 346.2)	158.3 (56.7, 331.0)	155.9 (61.7, 314.7)	153.4 (63.2, 290.9)	99.3 (38.6, 209.7)	103.7 (43.7, 207.7)
ST ^2^ (hours/week)	—	—	—	—	14.0 (7.0, 21.0)	14.0 (7.0, 21.0)	14.0 (9.0, 23.0)	19.5 (14.0, 28.0)	15.9 (9.0, 28.0)	14.0 (7.0, 25.3)
BMI ^1^ (kg/m^2^)	20.4 ± 1.7	20.6 ± 1.7	20.8 ± 1.8	21.0 ± 1.8	21.0 ± 1.8	21.1 ± 1.8	21.1 ± 1.9	21.2 ± 1.8	21.3 ± 1.8	21.4 ± 1.8
WC ^1^ (cm)		72.9 ± 6.7	74.2 ± 6.8	75.5 ± 7.3	76.3 ± 7.6	76.8 ± 7.4	77.6 ± 7.6	78.0 ± 8.2	77.6 ±10.4	79.2 ± 9.5

Abbreviations: Physical activity (PA), sedentary time (ST), body mass index (BMI), waist circumference (WC). ^1^ The value of this variable in the table is mean ± standard deviation. ^2^ The value of this variable in the table is median (interquartile range: Q1, Q3).

**Table 2 nutrients-14-00013-t002:** Baseline characteristics of the study population by multi-trajectories.

	Trajectories in Urban Areas					Trajectories in Rural Areas				
	IMP&H&LC	ABM	MP&VHF&VLC	MP&ILF&DHC	*p*	BM	MP&ILF&DHC	DMP&DHF&IMC	IMP&IHF&DLC	*p*
	(*n* = 1001, 38.7%)	(*n* = 857, 33.1%)	(*n* = 509, 19.7%)	(*n* = 219, 8.5%)		(*n* = 1870, 35.1%)	(*n* = 1135, 21.3%)	(*n* = 1071, 20.1%)	(*n* = 1252, 23.5%)	
Age ^1^ (years)	40.7 ± 15.3	45.5± 14.3	38.2 ± 14.6	51.9 ± 14.5	<0.001	40.4 ± 11.9	47.0 ± 14.5	39.4 ± 15.0	35.7 ± 12.4	<0.001
Men (%)	45.1	48.2	43.1	47.3	0.283	50.6	48.7	42.8	51.8	<0.001
North (%)	31.7	39.0	19.4	43.8	<0.001	36.5	47.8	24.8	29.7	<0.001
Education (%)	—	—	—	—	<0.001	—	—	—	—	<0.001
Primary and below	21.0	51.0	26.2	76.3		62.2	79.6	42.4	42.0	
Junior high	34.0	25.7	27.4	15.8		28.6	16.6	34.3	37.8	
Senior high and above	45.0	23.3	46.4	7.9		9.2	3.8	23.3	20.2	
Annual per capita family income (%)					<0.001					<0.001
Low (<10,000 RMB)	53.2	72.1	53.8	85.7		80.6	87.1	64.0	70.9	
Middle (10,000–20,000 RMB)	22.0	14.3	22.1	9.8		10.4	6.9	19.2	15.4	
High (>20,000 RMB)	24.8	13.6	24.1	4.4		9.0	6.0	16.8	13.7	
Smoker ^2^ (%)	29.5	35.8	31.1	35.5	0.020	38.3	37.6	31.2	35.0	<0.001
Alcohol drinker ^3^ (%)	35.6	39.9	35.1	47.7	0.004	36.3	35.6	32.4	33.9	0.136
Protein1 (%)	13.9 ± 3.1	12.1 ± 2.3	12.4 ± 3.0	11.8 ± 1.9	<0.001	11.7 ± 2.4	11.6 ± 1.8	12.7 ± 3.2	12.0 ± 2.7	<0.001
Fat ^1^ (%)	32.7 ± 9.3	27.1 ± 9.7	42.2 ± 10.2	17.3± 10.0	<0.001	23.0 ± 10.2	14.1 ± 7.4	34.1 ± 12.1	26.6 ± 11.8	<0.001
Carbohydrate ^1^ (%)	52.3 ± 9.9	59.9 ± 10.0	43.9 ± 10.1	69.7 ± 9.9	<0.001	64.4 ± 11.0	73.9 ± 7.8	52.1 ± 12.5	60.1 ± 13.0	<0.001
Energy intake ^1^ (kcal/d)	2232.2 ± 710.7	2293.0 ± 668.9	2408.8 ± 772.8	2,331.1 ± 708.6	<0.001	2422.5 ± 722.1	2493.0 ± 745.4	2282.8 ± 753.4	2345.4 ± 674.3	<0.001
PA ^4^ (METs/week)	146.8 (92.5,241.8)	251.1 (109.3, 519.9)	152.3 (88.7, 269.9)	439.8 (168.0, 667.5)	<0.001	475.5 (270.0, 656.0)	530.6 (348.0, 690.6)	270.2 (118.0, 480.3)	340.5 (159.6, 545.6)	<0.001
ST ^4^ (hours/week)	21.0 (14.0, 30.5)	16.0 (8.9, 24.5)	23.0 (14.0, 33.0)	9.0 (0.0, 18.5)	<0.001	14.0 (7.0, 21.0)	9.0 (3.5, 15.0)	14.0 (7.0, 23.0)	14.0 (7.0, 21.0)	<0.001
BMI ^1^ (kg/m^2^)	20.7 ± 1.8	20.9 ± 1.8	20.7 ± 1.8	20.8 ±1.6	0.139	20.7 ± 1.7	20.6 ± 1.7	20.5 ± 1.8	20.6 ± 1.7	0.060
WC ^1^ (cm)	74.9 ± 8.1	75.4 ± 7.5	74.4 ± 7.8	74.6 ±7.5	0.131	73.6 ± 6.6	74.1 ± 7.1	73.8 ± 7.1	73.4 ± 7.0	0.089

Abbreviations: Physical activity (PA), sedentary time (ST), body mass index (BMI), waist circumference (WC). ^1^ The value of this variable in the table is mean ± standard deviation. ^2^ Smoking status data were missing for 6 and 27 participants in urban and rural populations, respectively. ^3^ Alcohol drinking data were missing for 26 and 51 participants in urban and rural populations, respectively. ^4^ The value of this variable in the table is median (interquartile range: Q1, Q3).

**Table 3 nutrients-14-00013-t003:** Adjusted HRs and 95% CIs for risk of new-onset obesity according to multi-trajectories, 1991–2018.

Trajectories	Model 1 ^1^		Model 2 ^2^		Model 3 ^3^	
	*p* Value	*HR* (95% *CI*)	*p* Value	*HR* (95% *CI*)	*p* Value	*HR* (95% *CI*)
Urban trajectories						
IMP&HF&LC (versus (vs.) ABM)	0.605	0.85 (0.45, 1.59)	0.532	0.79 (0.38, 1.64)	0.671	0.85 (0.41, 1.79)
MP&VHF&VLC (vs. ABM)	0.945	0.97 (0.41, 2.31)	0.915	1.05 (0.40, 2.78)	0.912	1.06 (0.39, 2.85)
MP&ILF&DHC (vs. ABM)	0.423	0.65 (0.22, 1.88)	0.798	0.86 (0.28, 2.66)	0.709	0.80 (0.26, 2.52)
Rural trajectories						
MP&ILF&DHC (vs. BM)	0.681	1.08 (0.74, 1.57)	0.716	1.09 (0.69, 1.73)	0.948	0.98 (0.61, 1.58)
DMP&DHF&IMC (vs. BM)	0.168	0.72 (0.45, 1.15)	0.106	0.62 (0.35, 1.11)	0.034	0.50 (0.27, 0.95)
IMP&IHF&DLC (vs. BM)	0.052	0.67 (0.45, 1.00)	0.017	0.53 (0.32, 0.89)	0.008	0.48 (0.28, 0.83)

^1^ Model 1 is adjusted by age, gender, residence in north or south region, education, and family income. ^2^ Model 2 is further adjusted by alcohol consumption, cigarette smoking, fruit intake, vegetable intake, PA, ST, and sleep time. ^3^ Model 3 is further adjusted by baseline BMI, WC, total energy intake, and disease history (including diabetes, myocardial infarction, and stroke).

## Data Availability

Data sharing is not applicable to this article.

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
