# Peer review of "Multi-Trajectories of Macronutrient Intake and Their Associations with Obesity among Chinese Adults from 1991 to 2018: A Prospective Study"

_nutrients, 2021, doi:10.3390/nu14010013_

Round 1
Reviewer 1 Report
Authors described about the intake of different diet combinations (based on the quantity of proteins, carbohydrates and fats in the diet) and their association with developing obesity among urban and rural Chinese adults. Authors claim that low carbohydrate, high fat intake in rural population are associated with lower risk of obesity. The manuscript is well written and clear.
Author Response
Point 1: Authors described about the intake of different diet combinations (based on the quantity of proteins, carbohydrates and fats in the diet) and their association with developing obesity among urban and rural Chinese adults. Authors claim that low carbohydrate, high fat intake in rural population are associated with lower risk of obesity. The manuscript is well written and clear. 

Response 1: Thank you for taking time to review my manuscript.
Reviewer 2 Report
This study is a significant study that investigated the relationship between macronutrient intake and obesity in Chinese adults from 1991 to 2018.
The introduction and methods sections are also appropriately written.
However, the content of the dietary survey was omitted because it was written in a previous study, please describe the calculation method of nutrients, even if it is simple.
As you may have mentioned in your previous study, dietary content changes with the seasons.
When was the time of year that the study was conducted? If you have studied the effect of season on several measurements, please mention that as well.
Although the characteristics of the study population were statistically processed for both sexes, the proportion of males in the MP&ILF&DHC was significantly high in rural areas. In general, the ratio of fat to energy is higher for women. The odds ratio is addjusted for sex, but there is a possibility that it is not fully addjusted. Please add whether the results would have been the same if the analysis had been conducted separately for men and women.
Author Response
Point 1: This study is a significant study that investigated the relationship between macronutrient intake and obesity in Chinese adults from 1991 to 2018. The introduction and methods sections are also appropriately written. However, the content of the dietary survey was omitted because it was written in a previous study, please describe the calculation method of nutrients, even if it is simple. 

Response 1: The content of the dietary survey is further supplemented on the original basis, and the calculation method of macronutrient energy supply ratio is also given in detail. For details, see “2.3 Exposure Variables” of the revised manuscript.
Point 2: As you may have mentioned in your previous study, dietary content changes with the seasons.When was the time of year that the study was conducted? If you have studied the effect of season on several measurements, please mention that as well.
Response 2: The field work of each round was launched in autumn, considering the comparability between survey rounds. Thanks for your reminding, I have added this content to “2.3 Exposure Variables” of the revised manuscript.
Point 3: Although the characteristics of the study population were statistically processed for both sexes, the proportion of males in the MP&ILF&DHC was significantly high in rural areas. In general, the ratio of fat to energy is higher for women. The odds ratio is addjusted for sex, but there is a possibility that it is not fully addjusted. Please add whether the results would have been the same if the analysis had been conducted separately for men and women.
Response 3: There are indeed differences in the gender composition of different multi-trajectories in rural areas, so I included gender as a adjusting factor in the COX regression analysis. In the preliminary data analysis stage, I also tried to see the trajectories of different genders, but the results showed that there was almost no difference in the trajectories of macronutrient intake of different genders. In the stratification of urban and rural areas, it is found that the trajectory of urban and rural areas is completely different, so this manuscript only stratified urban and rural areas.